

# Self-reported side effects of COVID-19 vaccines among health professions students in India

Md Anwarul Azim Majumder[1,2], Ambadasu Bharatha[1], Santosh Kumar[3], Madhuri Chatterjee[4], Subir Gupta[1], Heather Harewood[1], Keerti Singh[1], WMS Johnson[5], Archana Rajasundaram[5], Sudeshna Banerjee Dutta[6], Sangishetti Vijay Prasad[7], Sayeeda Rahman[8], Russell Kabir[9], Ali Davod Parsa[9], Uma Gaur[1], Ahbab Mohammad Fazle Rabbi[10], Kandamaran Krishnamurthy[1], Shegufta Mohammad[11], Vikram Chode[12], Mainul Haque[13] and Michael H. Campbell[1]

[1] Faculty of Medical Sciences, The University of the West Indies, Cave Hill Campus, Bridgetown, Barbados
[2] Public Health Foundation of Bangladesh, Dhaka, Bangladesh
[3] Karnavati University, Ahmedabad, Gujarat, India
[4] Shri Ramkrishna Institute of Medical Science, Durgapur, West Bengal, India
[5] Sree Balaji Medical College and Hospital, Chennai, Tamil Nadu, India
[6] Department of Medical Surgical Nursing (Critical care nursing), Shri Anand Institute of Nursing, Rajkot, Gujarat, India
[7] Govt. Medical College, Shivpuri, Madhya Pradesh, India
[8] American University of Integrative Sciences (AUIS), Bridgetown, Barbados
[9] Anglia Ruskin University, Chelmsford, Essex, United Kingdom
[10] Department of Population Sciences, University of Dhaka, Dhaka, Bangladesh
[11] Health Education Department, Empower Me First College, Geneva, Switzerland
[12] Queen Elizabeth Hospital, Bridgetown, Barbados
[13] National Defence University of Malaysia, Kuala Lumpur, Malaysia

Corresponding authors
WMS Johnson, johnson-moses@gmail.com
Mainul Haque, mainul@upnm.edu.my

## ABSTRACT

Studies focusing on the safety and common side effects of vaccines play a crucial role in enhancing public acceptance of vaccination. Research is scarce regarding the usage of COVID-19 vaccines and the side effects experienced by health professions students in India and other countries. This study aimed to document self-reported side effects associated with COVID-19 vaccination among medical and dental students of six medical and dental colleges and teaching hospitals in four states (Tamil Nadu, Madhya Pradesh, Gujarat, and West Bengal) of India. A cross-sectional survey using purposive sampling of medical and dental students was conducted from 26 April to 26 May 2021. Data was collected using a Google Forms questionnaire capturing information regarding receiving COVID-19 vaccines, side effects and symptoms, onset and duration of symptoms, use of treatment to alleviate symptoms, awareness of haematologic risks associated with vaccination, and side effects from previous (non-COVID-19) vaccinations. The majority (94.5%) of participants received both doses of the Covishield/AstraZeneca COVID-19 vaccine. Among participants ($n = 492$), 45.3% ($n = 223$) reported one or more side effects. The most frequently reported side effects were soreness of the injected arm (80.3%), tiredness (78.5%), fever (71.3%), headache (64.1%), and hypersomnia (58.7%). The two most common severe symptoms were fever (14.8%) and headache (13%). Most side effects appeared on the day of vaccination:

soreness of the injection site (57%), fever (43.1%), and tiredness (42.6%). Most reported symptoms persisted for one to three days–soreness of the injection site (53%), fever (47.1%), and headache (42.6%). Logistic regression showed that women were almost 85% less likely to report side effects. The study's findings corroborate the safety of the Covishield/AstraZeneca vaccine's first dose, evidenced by the relatively minor and transient nature of the side effects. However, the study underscores the necessity for ongoing research to assess the long-term impacts of COVID-19 vaccines, especially in the context of booster doses, thereby contributing to the global understanding of vaccine safety and efficacy.

# INTRODUCTION

Coronavirus disease 19 (COVID-19) began as a local outbreak in Wuhan, China in late 2019 and was declared a pandemic in March 2020. The pandemic has caused approximately 7.6 million deaths globally through May 2023 (*World Health Organization, 2023a*). During the course of the pandemic, various COVID-19 control measures were applied in a layered manner, the so-called "Swiss cheese model" approach, as it became apparent that a combination of individual and population-based strategies were most effective in disrupting viral transmission (*Roberts, 2020*). Accordingly, initial reliance on non-pharmaceutical interventions (NPIs) such as handwashing and physical distancing was augmented in 2021 with the advent of COVID-19 vaccines. Concomitant with the relatively rapid introduction of vaccines, there has been a need to monitor for the possible emergence of side effects.

Importantly, the World Health Organization (WHO) had already listed vaccine hesitancy as a top-tier global health threat in 2019 (*World Health Organization, 2023b*). Accordingly, COVID-19 vaccine hesitancy has impeded uptake and threatened global containment efforts (*Alam et al., 2021*; *Krishnamurthy et al., 2021*; *MacDonald, 2015*; *Fedele et al., 2021*). Safety concerns were heightened due to accelerated vaccine trials to establish efficacy and safety before distribution (*Cole et al., 2022*). These concerns persisted despite good *in vivo* efficacy and safety profiles in Phase 1 to 3 trials (*Kaur et al., 2021a*) and emerging evidence supporting effectiveness in reducing disease transmission, severity, hospitalisations, and deaths (*Stokel-Walker, 2022*). Globally, COVID-19 vaccine hesitancy occurred both within the general population and among key sub-groups, including healthcare workers (*Alam et al., 2021*; *Krishnamurthy et al., 2021*; *Sallam, 2021*).

India has been severely affected by the COVID-19 pandemic. More than 45 million cases were detected and 0.53 million deaths were recorded (as of Jan 11, 2024) (*Worldometer, 2023*). In addition to non-pharmaceutical interventions, the Indian government launched the world's largest vaccination drive on January 16th, 2021, using two vaccines (ChAdOx1 nCoV-19 and BBV152) approved for emergency use and prioritizing healthcare workers (HCWs) for vaccination (*World Health Organization, 2021a*). Given India's

vast population, geographical range, cold chain requirements, and limited healthcare infrastructure, the comprehensive vaccination campaign was a monumental undertaking. These challenges notwithstanding, India implemented a vaccination programme that may be a useful model for developing nations (*Kumar et al., 2021*; *Pandey et al., 2021*). By May 27th, 2023, 67% of the Indian population was fully vaccinated (*Our World in Data, 2023*).

To boost acceptance and immunization rates, it is essential to continuously evaluate the safety of vaccinations and to provide timely and reliable evidence about side effect profiles (*Centers for Disease Control (CDC), 2022*; *World Health Organization, 2021b*). The WHO and Centers for Disease Control (CDC) have documented the rarity of severe side effects (*Centers for Disease Control (CDC), 2022*; *World Health Organization, 2021b*). Further studies have reported low rates of serious vaccine-related side effects in India (*Kaur et al., 2021b*; *Kamal et al., 2021*). Most recorded reactogenic symptoms were mild to moderate in strength, although a few were severe (*Kaur et al., 2021b*; *Kamal et al., 2021*; *Kundawar, Guralwar & Sontakke, 2022*; *Jayadevan, Shenoy & Anithadevi, 2021*; *Parida et al., 2022*; *Sharma, Jain & Vigarniya, 2022*). Pain at the injection site, swelling, and redness were the most common local side effects recorded. Systemic side effects included fever, exhaustion, myalgia, and headache. Most reactions were self-limiting and resolved within days. Healthcare providers have also experienced stress and depression after receiving the AstraZeneca vaccine. These professionals, already facing immense pressure during the COVID-19 pandemic, encountered additional stress from vaccine side effects. While the AstraZeneca vaccine is largely safe, individual reactions vary, potentially exacerbating stress in high-pressure work environments. Recent studies by *Al-Obaidy, Attash & Al-Qazaz (2022)* and *Madison et al. (2021)* highlighted potential stress and depression side effects of the AstraZeneca vaccine in healthcare providers. *Al-Obaidy, Attash & Al-Qazaz (2022)* reported increased rates of depression, anxiety, and stress post-vaccination among healthcare professionals. Their cross-sectional study did not compare post-vaccination symptoms to pre-vaccination baseline but did find an association between side effects and increased psychological symptoms. *Madison et al. (2021)* suggested that psychological states, like stress, may affect vaccine efficacy, indicating a complex relationship between mental health and vaccine response during the pandemic. Managing these effects requires a comprehensive approach, including encouraging healthcare workers to report side effects and providing them with mental health supports. addressing both physical and psychological impacts of vaccines in healthcare settings is essential.

Hesitancy among healthcare professionals frustrated governmental goals to prioritise vaccine administration for frontline healthcare workers to mitigate (*Alam et al., 2021*; *Krishnamurthy et al., 2021*; *Ashok et al., 2021*). Further, medical students' and healthcare workers' vaccine hesitancy contributed to vaccine hesitancy in the general population (*Alam et al., 2021*; *Peterson, Lee & Nugent, 2022*). Trepidation was exacerbated by conflicting information and misinformation regarding the motivation for vaccine development and anticipated negative externalities associated with vaccination (*Tyson, Johnson & Funk, 2020*). The spectre of the public health threat resulting from sub-optimal population coverage was particularly worrisome in those countries hardest hit by COVID-19, as well as those with limited health systems, like India. Concerns about efficacy, safety, and

convenience were the main hesitancy drivers among adult population (*Aliberti et al., 2022*). There is an urgent need for continuous public health efforts to maintain and increase trust in vaccines, especially by addressing safety concerns and providing clear, evidence-based information about the benefits and risks of vaccination.

In this context, health professions students are of particular interest. They represent an age cohort for which serious side effects are an acknowledged concern, and they may have an elevated risk of COVID-19 exposure during clinical training (*Bardosh et al., 2022*). We chose this group due to their unique status as future healthcare workers and their distinct demographic characteristics. The experiences of these students, typically young and healthy adults, offer valuable insights into milder vaccine side effects, which are crucial in understanding the vaccine's broader impact. The pandemic's alteration of their training from clinical to remote environments, and back, presents a unique stress and exposure scenario. By focusing on this group, we benefit from their medical knowledge and anticipated accurate self-reporting, enhancing our study's reliability. This specific focus adds depth to understanding vaccine side effects across various demographics, enriching the overall research landscape. The current study aimed to characterise the type and severity of self-reported side effects among health professions students in India after the introduction of COVID-19 vaccines in January 2021.

## MATERIAL AND METHODS

### Study design and participants

The survey used an instrument designed and employed by the research team in two previous studies (*Majumder et al., 2022*; *Majumder et al., 2023*). After validation, the questionnaire was administered using non-probability sampling between 26th April 2021 and 26th May 2021 and was distributed *via* Google Forms to students of six medical and dental colleges and teaching hospitals in four states (Tamil Nadu, Madhya Pradesh, Gujarat and West Bengal) of India.

Survey items solicited information on onset, type, severity, and persistence of common side effects after the first dose of COVID-19 vaccine, as well as demographic and vaccine information, and type of treatment (if any) to alleviate symptoms. Side effects were classified in terms of time of symptom *onset* (same day, 1–3 days' post-vaccination, 4–7 days' post-vaccination and none); *severity* (Severe–"I had to seek medical attention" Moderate–"I had to stop my daily activities"; Mild–"I was still able to do most daily activities"), and *duration* (1 day, 2–3 days, 4–7 days, still present).

### Data collection

Google Forms were anonymised by setting the software to not record IP addresses of the respondents. The survey link was shared *via* social media networks (Facebook, Messenger, WhatsApp, and Viber) and e-mail. Social media groups, professional associations, and healthcare organisations further assisted in distributing the survey to members. Participation was voluntary; the purpose of the study was explained, and participants were required to provide check-box consent to proceed to the online survey. No identifiable information was collected or stored.

### Ethical approval

Ethical approval was obtained from Institutional Review Board, Sree Balaji Medical College & Hospital, Bharath Institute of Higher Education and Research, Chennai, India (No:002/SBMCH/IHEC/2021/1178).

### Statistical analysis

Univariate measures (frequencies, percentages) were calculated to summarize demographic characteristics of participants and type and duration of vaccination-associated side effects. Bivariate analyses (chi-square) were performed to examine the relationship between demographic characteristics, existing comorbidities, and reported side effects. The individual effects of predictor variables on reactogenic symptoms were evaluated using binary logistic regression. All statistical analysis was conducted using IBM Statistical Package for Social Sciences (SPSS), Version 22.0 (IBM Corp., Armonk, NY, USA).

## RESULTS

### Respondents' characteristics

A total of 585 participants completed the questionnaire; of these, 492 had received at least one dose of vaccine. Most respondents were female (70.5%), undergraduate (96.9%), and dental students (57.4%). Just over half of respondents (56.1%) had received both two doses of COVID-19 vaccine at the time of the survey. Approximately 95% received the Covishield (AstraZeneca) vaccine.

Overall, 15% of the participants received COVID-19 vaccination. More unvaccinated respondents (39.1%) contracted the virus compared to persons who received two doses of vaccine (16.3%). Most respondents had no prior history of comorbidities (92.2%). Obesity (1.9%) and asthma (1.9%) were the most common comorbidities.

### Side effects following first dose of COVID-19 vaccine

The prevalence of vaccine-related side effects among respondents is shown in Fig. 1. More than 45.3% ($n = 223$) reported experiencing one or more side effects. The six most frequently reported side effects were: soreness of the injected arm (80.2%), tiredness (78.5%), fever (71.3%), headache (64.1%), hypersomnia (58.7%), and soreness of muscles (49.8%). The two most common severe symptoms were fever (14.8%), and headache (13%). Actions taken to alleviate vaccine-related symptoms included: paracetamol (38.8%), sleep (29.5%), and drinking water (22.6%). Only 11.6% of respondents had suffered similar side effects from previous vaccinations for other diseases (*e.g.*, Bacillus Calmette-Guérin-BCG and human papilloma virus-HPV). The respondents were aware of the risk of thromboembolic events (44.3%) and thrombocytopenia (34.1%), which occur as rare but serious complications (Table 1).

Severity, onset, and duration of the six most frequently reported side effects are summarized in Table 2. Most side effects appeared on the day of vaccination. Of these, the main complaints were soreness of the injected arm (57%), fever (43.1%), and tiredness (42.6%). Fever (23.8%) and tiredness (23.8%) were the most frequently reported symptoms appearing between day 1 to 3. Most reported symptoms persisted for 1 to 3 days. For 53% of
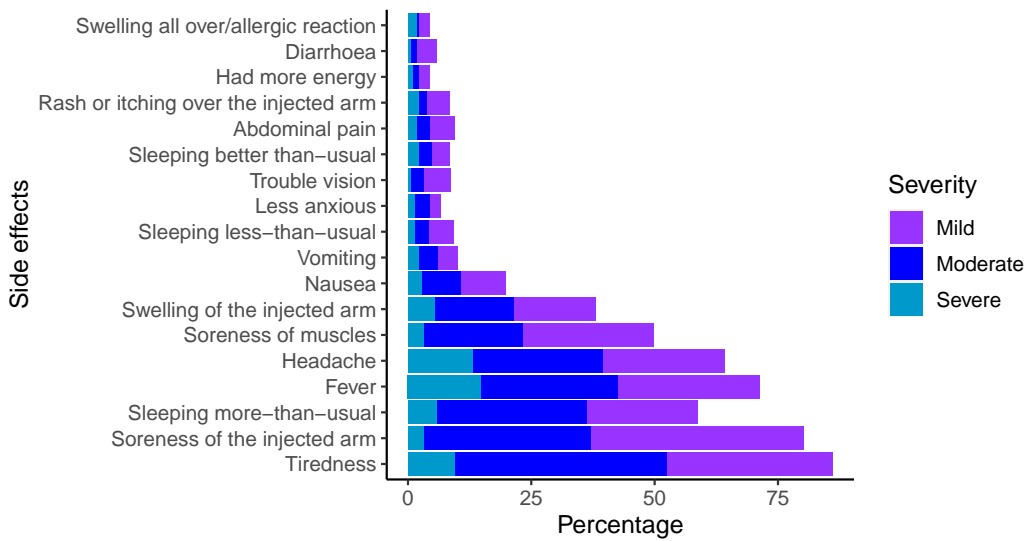

Figure 1 **Reported side effects of COVID-19 vaccination.**

participants, soreness in the injected arm lasted for 1 to 3 days, followed by fever (47.1%), and headache (42.6%). More persistent symptoms included soreness in the injected arm, which persisted 4 to 7 days for 16.1% of respondents and hypersomnia (2.3%) remaining for more than 7 days.

The prevalence of side effects among respondents stratified by gender, age, and field of study (medicine or dentistry) is summarized in Table 3. Sleep disturbance (hyper- and hyposomnia), increased energy, and swelling of the injected arm were significantly related to sex. Headache, swelling of the injected arm, and abdominal pain differed significantly by age group. Only feeling less anxious was significantly associated with participants' field of study.

### Determinants of side effects

Findings from the binary logistic regression model are summarized in Table 4. We evaluated six potential explanatory variables. COVID-19 test status and vaccine status were the only variables associated with the presence of symptoms. Symptomatic but never tested respondents were almost 85% less likely to report side effects than those who tested positive.

## DISCUSSION

Our study sought to document the side effects associated with COVID-19 vaccination among medical and dental students in India. Health professions students are a key demographic whose experiences with COVID-19 vaccination has not been studied on this scale in India previously. This group's experiences are vital for informing public health strategies, given their role as future healthcare providers and the increased vulnerability of healthcare workers to adverse mental health outcomes post COVID-19 vaccination.

**Table 1  Demographic and background characteristics of respondents.**

| Variables | Number | Percent |
|---|---|---|
| **Gender** ($n = 492$) | | |
| Male | 145 | 29.5 |
| Female | 347 | 70.5 |
| **Age (in years)** ($n = 492$) | | |
| ≤20 | 254 | 51.6 |
| 21–25 | 221 | 44.9 |
| 25+ | 17 | 3.5 |
| **Study level** ($n = 491$) | | |
| Undergraduate | 476 | 96.9 |
| Graduate | 15 | 3.1 |
| **Specific year of study level** ($n = 449$) | | |
| 1st year | 133 | 29.6 |
| 2nd year | 161 | 35.8 |
| 3rd year | 44 | 9.8 |
| 4th year | 51 | 11.3 |
| CRRI/Intern | 60 | 13.3 |
| **Specific field of study** ($n = 486$) | | |
| Medicine | 203 | 41.8 |
| Dentistry | 279 | 57.4 |
| Other | 4 | 0.8 |
| **Vaccination status** ($n = 492$) | | |
| 1st dose only | 216 | 43.9 |
| Both first and second doses | 276 | 56.1 |
| **Vaccine type** ($n = 492$) | | |
| Covishield (AstraZeneca) | 465 | 94.5 |
| Covaxin (India) | 27 | 5.5 |
| **COVID-19 test status** ($n = 492$) | | |
| Yes, tested positive (RT-PCR) | 71 | 14.4 |
| Yes, tested positive (CT) | 6 | 1.2 |
| Yes, never tested (symptomatic) | 16 | 3.3 |
| No | 399 | 81.1 |
| **Time of COVID-19 infection** ($n = 92$) | | |
| Before 1st dose | 36 | 39.1 |
| Between 1st and 2nd dose | 41 | 44.6 |
| After 2nd dose | 15 | 16.3 |
| **Comorbidities, if any** ($n = 474$) | | |
| No illness | 437 | 92.2 |
| Obesity | 9 | 1.9 |
| Asthma | 9 | 1.9 |
| Hypertension | 2 | 0.4 |
| Hypothyroidism | 2 | 0.4 |

**Table 1** (*continued*)

| Variables | Number | Percent |
|---|---|---|
| Autoimmune diseases | 1 | 0.2 |
| Diabetes and others | 3 | 0.6 |
| Others | 11 | 2.3 |
| **Occurrence of side effects due to COVID-19 vaccines** ($n = 492$) | | |
| Yes | 223 | 45.3 |
| No | 269 | 54.7 |
| **Awareness of increased risk of blood clots (thromboembolic events)** ($n = 492$) | | |
| Yes | 218 | 44.3 |
| No | 165 | 33.5 |
| Don't know | 109 | 22.2 |
| **Awareness of the increased risk of low platelets (thrombocytopenia)** ($n = 492$) | | |
| Yes | 168 | 34.1 |
| No | 208 | 42.3 |
| Don't know | 116 | 23.6 |
| **Side effects from previous vaccinations for other diseases (e.g., BCG, HPV)** | | |
| Yes | 57 | 11.6 |
| No | 183 | 37.2 |
| Couldn't remember | 252 | 51.2 |
| **Actions taken to alleviate vaccine-related symptoms** | | |
| Paracetamol | 191 | 38.8 |
| Sleep | 145 | 29.5 |
| Drinking water | 111 | 22.6 |
| Cold bath/shower/sponging | 27 | 5.5 |
| Other pain killer | 25 | 5.1 |
| Ibuprofen | 18 | 3.7 |
| Nothing taken | 15 | 3.0 |
| Nothing worked | 9 | 1.8 |
| Other actions taken | 8 | 1.6 |

**Notes.**

CRRI, Compulsory Rotatory Residential Internship; RT-PCR, Reverse transcription polymerase chain reaction; CT, Cycle Threshold.

Findings concerning side effects are crucial in a densely populated country heavily impacted by COVID-19 and also contribute to global understanding of vaccine safety, offering valuable data for similar demographics worldwide. Our study fills an important gap in knowledge about vaccine side effects, aiding effective policy and practice in India and beyond.

The COVID-19 pandemic inflicted enormous burden on economic, social, and healthcare infrastructure, disrupting lives and livelihoods (*Alam et al., 2021*; *Rahman et al., 2021*; *Habas et al., 2020*; *Azim Majumder & Razzaque, 2022*). In the absence of clearly effective treatments in the early phases of the pandemic, evidence-deficient interventions, including repurposed medications, nutraceuticals, complementary and alternative therapies, multiple vitamins, and immunity-promoting agents, were haphazardly utilized in many countries (*Charan et al., 2021a*; *Charan et al., 2021b*; *Dutta et al., 2021a*;

**Table 2  Prevalence of six most reported side effects (n = 492).**

| Side effects | The severity of side effects | | | | Onset | | | | Duration of symptoms | | | | |
|---|---|---|---|---|---|---|---|---|---|---|---|---|---|
| | Severe | Moderate | Mild | Total | Same day | 1–3 days | 4–7 days | Total | Same day | 1–3 days | 4–7days | >7 days | Total |
| Soreness of injected arm | 7 (3.14%) | 76 (34.08%) | 96 (43.05%) | 179 (80.27%) | 127 (56.95%) | 47 (21.08%) | 3 (1.35%) | 177 (79.37%) | 24 (10.76%) | 118 (52.91%) | 36 (16.14%) | 1 (0.44%) | 179 (8.27%) |
| Soreness of muscles | 7 (3.14%) | 45 (20.18%) | 59 (26.46%) | 111 (49.78%) | 65 (29.15%) | 43 (19.28%) | 3 (1.35%) | 111 (49.78%) | 11 (4.93%) | 85 (38.12%) | 15 (6.73%) | 5 (2.24%) | 116 (52.02%) |
| Fever | 33 (14.79%) | 62 (27.80%) | 64 (28.7%) | 159 (71.3%) | 96 (43.05%) | 53 (23.77%) | 0 (0%) | 149 (66.82%) | 36 (16.14%) | 105 (47.09%) | 7 (3.14%) | 3 (1.35%) | 151 (67.71%) |
| Headache | 29 (13%) | 59 (26.46%) | 55 (24.66%) | 143 (64.13%) | 88 (39.46%) | 46 (20.63%) | 1 (0.44%) | 135 (60.54%) | 33 (14.79%) | 95 (42.6%) | 7 (3.14%) | 4 (1.79%) | 139 (62.33%) |
| Tiredness | 21 (9.42%) | 79 (35.42%) | 75 (33.63%) | 175 (78.48%) | 95 (42.6%) | 53 (23.78%) | 4 (1.79%) | 152 (68.16%) | 30 (13.45%) | 92 (41.26%) | 16 (7.17%) | 3 (1.35%) | 141 (63.23%) |
| Hypersomnia | 13 (5.83%) | 68 (30.49%) | 50 (22.42%) | 131 (58.74%) | 66 (29.6%) | 42 (18.83%) | 4 (1.79%) | 112 (50.22%) | 29 (13%) | 68 (30.49%) | 5 (2.24%) | 6 (2.27%) | 108 (48.43%) |

**Table 3  Side effects among medical and dental students stratified by gender, age, and field of study.**

| Side effect | Gender | | | | Age | | | | Field of study | | | |
|---|---|---|---|---|---|---|---|---|---|---|---|---|
| | Male (n = 145) | Female (n = 347) | Total 492 | p-value | ≤20 years (n = 254) | >20 years (n = 238) | Total 492 | p-value | Medicine (n = 203) | Dentistry (n = 279) | Total 482 | p-value |
| Soreness of the injected arm | 42 | 137 | 179 | 0.346 | 88 | 91 | 179 | 0.334 | 76 | 100 | 176 | 0.834 |
| Soreness of muscles | 27 | 84 | 111 | 0.565 | 61 | 50 | 111 | 0.367 | 43 | 65 | 109 | 0.410 |
| Fever | 40 | 109 | 159 | 0.077 | 85 | 74 | 159 | 0.440 | 60 | 95 | 155 | 0.110 |
| Headache | 38 | 105 | 143 | 0.579 | 74 | 69 | 143 | **0.025** | 48 | 81 | 129 | 0.967 |
| Vision trouble | 7 | 12 | 19 | 0.102 | 11 | 8 | 19 | 0.137 | 12 | 7 | 19 | 0.125 |
| Tiredness | 44 | 131 | 175 | 0.717 | 106 | 79 | 185 | 0.519 | 72 | 98 | 170 | 0.982 |
| Hypersomnia | 26 | 105 | 131 | 0.247 | 72 | 59 | 131 | 0.125 | 56 | 72 | 128 | 0.218 |
| Hyposomnia | 12 | 9 | 21 | **0.001** | 10 | 11 | 21 | 0.693 | 10 | 10 | 20 | 0.084 |
| Sleeping more than usual | 10 | 9 | 19 | **0.024** | 12 | 7 | 19 | 0.644 | 7 | 12 | 19 | 0.063 |
| Had more energy | 7 | 3 | 10 | **0.020** | 5 | 5 | 10 | 0.966 | 6 | 4 | 10 | 0.265 |
| Less anxious | 6 | 7 | 13 | 0.383 | 8 | 7 | 15 | 0.331 | 6 | 9 | 15 | **0.023** |
| Swelling of injected arm | 12 | 73 | 85 | **0.000** | 48 | 37 | 85 | **0.030** | 29 | 54 | 83 | 0.0.29 |
| Swelling all over/allergic reaction | 3 | 7 | 10 | 0.253 | 4 | 6 | 10 | 0.387 | 4 | 5 | 9 | 0.649 |
| Rash/itching on injected arm | 6 | 13 | 19 | 0.955 | 9 | 10 | 19 | 0.848 | 6 | 12 | 18 | 0.557 |
| Abdominal pain | 6 | 15 | 21 | 0.221 | 11 | 10 | 21 | **0.026** | 8 | 12 | 20 | 0.183 |
| Diarrhea | 5 | 8 | 13 | 0.116 | 7 | 6 | 13 | 0.053 | 5 | 6 | 11 | 0.547 |
| Nausea | 7 | 32 | 39 | 0.051 | 23 | 21 | 44 | 0.062 | 22 | 22 | 44 | 0.200 |
| Vomiting | 3 | 20 | 23 | 0.226 | 15 | 8 | 23 | 0.090 | 7 | 16 | 23 | 0.086 |

**Notes.**
Bold font indicates statistical significance: $p \leq 0.05$.

*Kaur et al., 2020*; *Samad et al., 2021a*; *Samad et al., 2021b*; *Hossen et al., 2020*). According to the Regulatory Affairs Professionals Society, as of January 27, 2023, there were 97 vaccine candidates under development and 37 approved COVID-19 vaccines globally (*Craven, 2023*). From the first deployment of COVID-19 vaccines, vaccination campaigns prioritized healthcare workers given their essential function and elevated risk of exposure. As with most pharmaceuticals, COVID-19 vaccine-associated side effects were reported

**Table 4   Logistic regression coefficients and odds ratios (95% CI) for determinants of vaccine side effects.**

| Variables | Odds ratio (OR) | 95% CI | *p*-value |
|---|---|---|---|
| **Gender of respondent** <br> Male (Ref) <br> Female | 0.783 | 0.253, 2.247 | 0.672 |
| **Respondent's field of study** <br> Medicine (Ref) <br> Dentistry | 1.007 | 0.317, 3.197 | 0.991 |
| **Vaccination Status** <br> First dose only (Ref) <br> Both first and second doses | 0.326 | 0.104, 1.016 | 0.053 |
| **COVID-19 test status** <br> Yes, tested positive (RT-PCR) (Ref) <br> Yes, tested positive (CT) <br> Never tested (symptomatic) <br> No | 1.167 <br> 0.152 <br> 0.466 | 0.131, 10.404 <br> 0.027, 0.865 <br> 0.075, 2.907 | 0.890 <br> 0.034 <br> 0.413 |
| **Prior presence of any chronic illness** <br> No illness (Ref) <br> Presence of illness | 1.155 | 0.231, 5.784 | 0.861 |
| **Time of COVID infection** <br> Before first dose (Ref) <br> Between two doses <br> After 2nd dose | 1.921 <br> 1.095 | 0.633, 5.833 <br> 0.226, 5.307 | 0.249 <br> 0.911 |

**-2log likelihood:** 105.218

**Cox & Snell R Square:** 0.148

**Nagelkerke R Square:** 0.198

globally and monitored using the WHO pharmacovigilance database, igiBase (*Kaur et al., 2021a*; *Dutta et al., 2022*; *Jeet Kaur et al., 2021*; *Dutta et al., 2021b*).

We investigated the prevalence, onset, duration, and severity of self-reported side effects among students, most of whom (95%) had received Covishield vaccines. In two studies conducted in India and Bangladesh, the percentages of participants who received Covishield vaccines were 91% (*Majumder et al., 2023*) and 100%, (*Majumder et al., 2022*) respectively. In the present study, less than half (45.3%) of respondents reported one or more vaccine-related side effects after first doses of Covishield vaccines. This level of side effects aligns with global reports from similar demographics, suggesting a pattern in the immunological response to the COVID-19 vaccines that transcends geographical boundaries. However, the severity and perception of these side effects could be influenced by the participants' medical knowledge, potentially leading to underreporting or over reporting. Two studies conducted in India found that over 40% of healthcare workers experienced at least one side effect after the first dose of vaccine (*Kaur et al., 2021b*; *Majumder et al., 2023*). However, other studies conducted among healthcare workers in India reported higher rates of side effects: *Kamal et al. (2021)* (57%), *Kundawar, Guralwar & Sontakke (2022)* (68.4%), *Kataria et al. (2022)* (69.7%), *Jayadevan, Shenoy & Anithadevi (2021)* (66%). Two studies conducted in Nepal found especially high side effect rates compared to the other studies: 91.6% (*Gautam et al., 2021*) and 85% (*Shrijana Shrestha et al., 2021*).

The three most frequently reported side effects were soreness of the injected arm, tiredness, and fever. These findings are consistent with those of our previous surveys of healthcare workers in Bangladesh (*Majumder et al., 2022*) and India (*Majumder et al., 2023*), as well as other Indian studies reporting pain or tenderness at the injection site as very common vaccine-associated side effects (*Kaur et al., 2021b*; *Parida et al., 2022*; *Sharma, Jain & Vigarniya, 2022*; *Jhaj et al., 2022*). Tiredness and fever were also frequently reported, consistent with data from the electronic medicines compendium (*Electronic medicines compendium, 2023*). Several studies (*Jayadevan, Shenoy & Anithadevi, 2021*; *Sharma, Jain & Vigarniya, 2022*; *Shrijana Shrestha et al., 2021*; *Mahapatra et al., 2021*) found that malaise/tiredness and fever were commonly reported symptoms, although other researchers reported less frequent incidence of fever and malaise in comparison to the current study (*Kaur et al., 2021b*; *Kundawar, Guralwar & Sontakke, 2022*; *Jayadevan, Shenoy & Ts, 2021*; *Sharma, Jain & Vigarniya, 2022*; *Shrijana Shrestha et al., 2021*; *Jhaj et al., 2022*; *Mahapatra et al., 2021*).

Research on the use of the Covishield vaccine among health professions students in India and other countries is limited. A study in India (*Nandini et al., 2022*) found that 71% of students reported side effects after the first vaccination, a rate considerably higher than our findings. The most common side effects were pain at the site of injection (74.2%), fever (40.4%), body pain (26%), and headache (21.6%). In another Indian study (*Peradi et al., 2022*) of healthcare workers and students at a dental college, 70% of respondents reported one or more side effects–injection site pain (60.8%), weakness (60.8%), and fever (60%).

Our results indicate that respondents experienced side effects primarily on the day of vaccination, with more than half of the participants (56.95%) reporting soreness in the arm, followed by fever (43.0%) and tiredness (42.6%). Studies conducted in India (*Kaur et al., 2021b*; *Majumder et al., 2023*; *Jhaj et al., 2022*) and Bangladesh (*Majumder et al., 2022*) found similar patterns of symptoms. Approximately half of the physicians and dentists reported soreness in the arm (48.4%), and nearly one-third reported fever (31.9%) and headache (31.9%) on the day they were vaccinated. In the Bangladeshi study, same-day soreness in the arm and fever were prominent symptoms, reported by 46.5% and 34.3% of physicians respectively.

Respondents in the present study reported fewer side effects between days 1–3 and the least on days 4–7. Most side effects persisted for one to three days and ameliorated by day 7, indicating that symptoms were usually short-lived and did not significantly impact daily activities. This is consistent with previously published findings on symptom presentation and severity (*Kamal et al., 2021*; *Sharma, Jain & Vigarniya, 2022*; *Majumder et al., 2022*; *Majumder et al., 2023*; *Kaur et al., 2022*). *Sharma, Jain & Vigarniya (2022)* reported that side effects after vaccination persisted for only hours for most respondents (63%), and only 9% reported symptoms persisting for 3–7 days. Studies by *Kaur et al. (2021b)*, *Kamal et al. (2021)*, and *Kaur et al. (2022)* reported similar presentations of side effects after vaccination. Additional evidence suggests that side effects are more common after the first than after subsequent doses of vaccine (*Kaur et al., 2022*; *Hatmal et al., 2021*; *Menni et al., 2021*). The WHO and CDC both report that side effects associated with COVID-19 vaccines are generally mild to moderate and tend to remit within a few days. Thus, the

substantial and growing body of findings supporting predominantly manageable severity and limited duration of side effects is important to address safety concerns that may act as barriers to vaccine uptake.

Previous studies have shown that post vaccination side effects were more common among women than men (*Kaur et al., 2021b*; *Jayadevan, Shenoy & Anithadevi, 2021*; *Menni et al., 2021*). Interestingly, our analysis revealed that females were more prone to develop certain side effects, such as swelling in the injected arm, a finding that echoes the gender-specific reactogenic patterns noted in other studies (*Nandini et al., 2022*). We found that female medical students in South India were more prone to develop swelling in the injected arm. In a Bangladeshi study, the more frequently reported side effects—fever, visual disturbance, increased sleep, rash or itching near the injection site, and nausea were significantly more common among females (*Majumder et al., 2022*). However, study in India found that female health workers were almost 60% less likely to report side effects (*Majumder et al., 2023*).

The study's findings on the self-management strategies employed by health professions students provide insights into how medical knowledge might influence responses to side effects. This is a critical aspect that differentiates our study group from the general population, as it suggests a higher degree of health literacy and self-management capabilities among health professions students. Some respondents who experienced vaccine-related side effects used measures such as paracetamol, sleeping, and drinking water. These common self-care measures for managing mild to moderate vaccine adverse reactions indicate that participants actively managed vaccine side effects, as expected given their healthcare training. It is also interesting to note that some respondents had experienced similar side effects from previous vaccinations for other diseases. This suggests that adverse reactions to vaccines may not be solely vaccine-specific and may reflect more a general propensity for reactogenicity. The likelihood of reactogenic responses depends on a combination of host factors (including, age, gender, pre-existing immunity), vaccine characteristics, and/or vaccine adjuvants (*Hervé et al., 2019*). The use of painkillers was also documented other studies (*Jayadevan, Shenoy & Anithadevi, 2021*; *Sharma, Jain & Vigarniya, 2022*; *Majumder et al., 2022*; *Majumder et al., 2023*; *Kaur et al., 2022*; *Hatmal et al., 2021*). These findings could inform strategies to educate the broader population on managing common side effects.

Additionally, the findings indicate that respondents were aware of the rare complications that can occur with the COVID-19 vaccine, such as thromboembolic events and thrombocytopenia. This may indicate that vaccine safety messaging is reaching the public, and heightened awareness could contribute to better preparedness and vigilance for vaccine-related safety issues and response to vaccine-related adverse events, enhancing the overall safety profile of vaccination campaign (*OECD, 2021*).

The finding that most of the COVID-19-positive cases contracted the virus before receiving a first vaccination highlights the importance of early vaccination among naive populations. Prior studies have demonstrated that vaccination decreases the risk of COVID-19 infection and minimizes the risk of complications (*Ashok et al., 2021*; *Habas et al., 2020*; *Azim Majumder & Razzaque, 2022*). *Menni et al. (2021)* reported that

ChAdOx1 vaccination reduced infection by 60% [95% CI 68% to 49%] at 21–44 days post-vaccination. Further studies have demonstrated that vaccinated persons are less likely to develop symptoms, experience severe disease (or require hospitalization), develop complications, or transmit COVID-19 to others (*Hatmal et al., 2021*; *Menni et al., 2021*; *Antonelli et al., 2022*; *Prosser Scully, 2022*; *Richterman, Meyerowitz & Cevik, 2021*; *Flacco et al., 2022*).

The study's findings regarding self-management strategies employed by health professions students provide insights into how medical knowledge might influence responses to side effects. These findings could inform educational efforts for the broader population on managing common side effects. This study augments evidence supporting public health recommendations for COVID-19 vaccination as a major plank in the containment of the disease. Professional and public support are crucial to sustain future vaccination campaigns that may be needed, given that the future course of COVID-19 or similar emerging threats is not certain. Clear and accurate information regarding COVID-19 vaccination provided by governments through multifaceted outreach programmes to both the general public and to key health system populations is vital to boost vaccine confidence and uptake.

### Strengths and limitations of the study

The presumed ability of students in health professions to accurately identify and report symptoms is a strength of this study. Important limitations to generalizability of findings include the relatively small sample size, voluntary response sampling, and cross-sectional design and the focus on short-term side effects. The study's reliance on self-reported data render the findings subject to recall and reporting bias. The accuracy of the data depends on the participants' ability to adequately remember and report their experiences, which can be influenced by various factors including their perception of the severity of the side effects. Further, cultural factors influencing perception and reporting of symptoms limit the utility of findings from India in other settings.

## CONCLUSION

Our findings demonstrated that most medical and dental students who received the COVID-19 vaccination experienced primarily mild to moderate symptoms, with severe side effects rarely reported. Soreness at the injected arm (80.2%) and tiredness (78.5%) were the commonest reported side effect. Non-probability sampling is an important limitation of this study as it constrains generalizability of the findings. Future studies using probability sampling and longitudinal design are needed to elucidate potential longer-term side effects of COVID-19 vaccines. In summary, this study's findings augment the growing body of evidence supporting the safety and tolerability of COVID-19 vaccines. They contribute significantly to the discourse on vaccine safety, particularly in the context of health professions students, and underscore the importance of ongoing surveillance and research to ensure vaccine safety and efficacy.

## ACKNOWLEDGEMENTS

The authors express their gratitude to the students who willingly participated in this study and diligently completed the questionnaire.

### Funding

The authors received no funding for this work.

### Competing Interests

Russell Kabir is an Academic Editor for PeerJ. The other authors declare that they have no competing interests.

### Author Contributions

- Md Anwarul Azim Majumder conceived and designed the experiments, analyzed the data, prepared figures and/or tables, authored or reviewed drafts of the article, lead and coordinate the research team,, and approved the final draft.
- Ambadasu Bharatha conceived and designed the experiments, analyzed the data, prepared figures and/or tables, authored or reviewed drafts of the article, and approved the final draft.
- Santosh Kumar performed the experiments, authored or reviewed drafts of the article, and approved the final draft.
- Madhuri Chatterjee performed the experiments, authored or reviewed drafts of the article, and approved the final draft.
- Subir Gupta analyzed the data, prepared figures and/or tables, authored or reviewed drafts of the article, and approved the final draft.
- Heather Harewood analyzed the data, prepared figures and/or tables, authored or reviewed drafts of the article, corrected the final version for English and Grammar, and approved the final draft.
- Keerti Singh conceived and designed the experiments, authored or reviewed drafts of the article, and approved the final draft.
- WMS Johnson conceived and designed the experiments, performed the experiments, authored or reviewed drafts of the article, worked for IRB application, and approved the final draft.
- Archana Rajasundaram conceived and designed the experiments, performed the experiments, authored or reviewed drafts of the article, worked for IRB application, and approved the final draft.
- Sudeshna Banerjee Dutta performed the experiments, authored or reviewed drafts of the article, and approved the final draft.
- Sangishetti Vijay Prasad performed the experiments, authored or reviewed drafts of the article, and approved the final draft.

- Sayeeda Rahman conceived and designed the experiments, analyzed the data, prepared figures and/or tables, authored or reviewed drafts of the article, and approved the final draft.
- Russell Kabir analyzed the data, prepared figures and/or tables, authored or reviewed drafts of the article, and approved the final draft.
- Ali Davod Parsa analyzed the data, prepared figures and/or tables, authored or reviewed drafts of the article, and approved the final draft.
- Uma Gaur analyzed the data, authored or reviewed drafts of the article, and approved the final draft.
- Ahbab Mohammad Fazle Rabbi analyzed the data, prepared figures and/or tables, authored or reviewed drafts of the article, did detailed data analysis, and approved the final draft.
- Kandamaran Krishnamurthy conceived and designed the experiments, authored or reviewed drafts of the article, and approved the final draft.
- Shegufta Mohammad conceived and designed the experiments, authored or reviewed drafts of the article, and approved the final draft.
- Vikram Chode analyzed the data, prepared figures and/or tables, authored or reviewed drafts of the article, and approved the final draft.
- Mainul Haque conceived and designed the experiments, authored or reviewed drafts of the article, and approved the final draft.
- Michael H. Campbell analyzed the data, prepared figures and/or tables, authored or reviewed drafts of the article, corrected the final version for English and Grammar, and approved the final draft.

## Human Ethics

The following information was supplied relating to ethical approvals (*i.e.*, approving body and any reference numbers):

Institutional Review Board, Sree Balaji Medical College & Hospital, Bharath Institute of Higher Education and Research, Chennai, India (No:002/SBMCH/IHEC/2021/1178).

## Data Availability

The raw data are available in the Supplementary File.

## Supplemental Information

Supplemental information for this article can be found online at http://dx.doi.org/10.7717/peerj.17083#supplemental-information.

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
