# Peer review of "Self-reported side effects of COVID-19 vaccines among health professions students in India"

_PeerJ, doi:10.7717/peerj.17083_

## Round 0.1 · original submission · Minor Revisions

The manuscript is generally well structured. However, you should take into consideration all reviewers' comments to improve its quality.

The introduction of the abstract repeat exactly the same sentence of the introduction.Try to revise it.

Try to improve the quality of the figure and the tables

Reviewer 1 ·

Basic reporting

Please see the attachment.

Experimental design

Please see the attachment.

Validity of the findings

Please see the attachment.

Additional comments

Please see the attachment.

** Annotated reviews are not available for download in order to protect the identity of reviewers who chose to remain anonymous.

Reviewer 2 ·

Basic reporting

TThe article is written in good, clear English. The number of references needs to be increased. The article is organized in an acceptable format and the figures are relevant to the article. Coherent body of work.

Experimental design

The research is within the scope of the journal. The research questions are defined, but it is unclear whether there was actually a gap in this part of the research. The research was conducted according to ethical standards. The methods section needs improvement.

Validity of the findings

The study has moderate impact and novelty. It can be considered a replication study. The data need improvement. The conclusions need improvement.

Additional comments

Comments on Majumder et al. Self-reported side effects of COVID-19 vaccines among health professional students in India


It is an interesting study investigating the side effects of Covid-19 vaccines among health professional students, but the paper needs improvement.

Introduction

There are also other scientific papers, not cited, that highlight vaccine side effects and hesitations. For example, Li M. et al. who analyze the hesitations and related factors towards the covid vaccination of medical students; or Aliberti S.M. et al. who show side effects, attitudes, and hesitations towards the covid Vaxzervria vaccination, moreover the approach and settings of your study seem very similar to theirs.

Please create an appropriate and larger publication unit.

Material and Methods
Study design and participants
How was the questionnaire validated? Please explain.
Please delete the paragraph beginning with the words "Survey items solicited..." and ending with "still present" from line 133-139 from the Study Design and Participants section and insert it in the Data Collection section.

Data collection
Please indicate the link to the survey and the date of last access or when it was discontinued.

Statistical analysis
Please explain better what measures were used to analyze the data so that the study can be more understandable. For example, which data are dichotomized, which are continuous, etc.

Results
Determinants of side effects
Please explain the most important results of Table 4 related to logistic regression
In Table 4, variable definition section: please write first the variable description column; second column write OR; third column 95% CI; fourth column p value. Also, write what is the log likelihood of the model analyzed, and better highlight which are the outcome variables compared to the explanatory variables. The table is unclear.

Discussion
Please remove the percentages from the discussion and insert them into the results. In the discussion section, describe the most important data from the study and compare it to other studies. Avoid always writing the results of your two previous studies first and then the results of the current study. Limit the use of the words "our previous studies. Instead, show what is new in this study compared to previous studies. Write the discussion in a better way.

Reviewer 3 ·

Basic reporting

Clear and professional English has been used, however, there are many grammatical errors that need to be corrected.

Experimental design

Study Aim and Idea: Although it is important to know the side effects of COVID-19 vaccines this is not a novel area of research and these side effects have been identified and are being updated on a regular basis. In addition, the side effects of COVID-19 won't change if these are medical or dental students hence the selection of participants does not make sense. These side effects have already been identified as part of several clinical trials.

Method: The authors describe the survey as validated but there is no information provided on any kind of survey validation.

Validity of the findings

The results cannot be validated as the study was not conducted in a controlled environment and was based on a recall basis.

Reviewer 4 ·

Basic reporting

• Your paper is well-written with a simple message. However, some issues need to be addressed, including more details about the AstraZeneca vaccine. What about other types of adverse effects? Does the vaccine cause stress and depression? several pieces of research from different perspectives studied whether the vaccine causes any psychological effects. I would suggest that some of these studies could be added to the introduction. See the citations below.

1. Al-Obaidy LM, Attash HM, Al-Qazaz HK. Depression, anxiety and stress after COVID-19 vaccination: A retrospective cross-sectional study among health care providers. Pharmacy Practice. 2022 Sep 30;20(3):1-8.
2. Madison AA, Shrout MR, Renna ME, et al. Psychological and Behavioral Predictors of Vaccine Efficacy: Considerations for COVID-19. Perspect Psychol Sci. 2021;16(2):191-203. https://doi.org/10.1177/174569162198924313.

Experimental design

'no comment'

Validity of the findings

The paper describes interesting data drawn from a cross-sectional survey of the adverse effects of COVID-19 vaccines.
In my opinion, the paper could provide a valuable addition to the literature and is publishable. Well done and I enjoyed reading this paper.
The authors have identified the evidence of their conclusions in the manuscript appropriately. I could not identify any further conceptual limitations in the research presented. Overall, the research has been conducted and presented well.

Additional comments

• Change the title (Materials and Methods) to (Patients and Methods).
• Although the abbreviation (SPSS) is a known term, it must be written in the full name to be understood by the reader (Statistical Package for Social Sciences).
Line 180, write the full term of the abbreviation (HPV) to be more understandable for the reader.
HPV: Human Papilloma Virus
• The discussion appeared rather long. A better-organized discussion section would also strengthen your manuscript.
Delete reference no 60 as it is repeated (ref no. 56)
• Ref. no 66: ‘’………Available online: 571 https://www.who.int/news-room/feature-stories/detail/side-effects-of-covid-19- 572 vaccines (accessed on ……..?)’’. Add the date of access.
• For the website references, always include the date you accessed the site at the end of the reference (which includes: day, month, and year), so add the year for the following web references: (Ref. no. 1, 2, 12, 13, 16, 17, 18, 27, 41, 51, 67, and 68).

---

## Round 0.2 · accepted · Accept

The authors have answered to all Editor and reviewers comments.

Reviewer 1 ·

Basic reporting

No further comments.

Experimental design

No further comments.

Validity of the findings

No further comments.

Additional comments

No further comments.

Reviewer 2 ·

Basic reporting

The authors have done a good job of reviewing the manuscript. I agree to publish the manuscript.

The article is written in good, clear English. Literature references is sufficiently.

Experimental design

The research is well defined and meaningful.

Validity of the findings

All underlying data have been provided.

Additional comments

I wish much success with this publication.

Reviewer 4 ·

Basic reporting

Thank you for the opportunity to review the revised manuscript, entitled " Self-reported side effects of COVID-19 vaccines among health professions students in India’’
The authors conducted a cross-sectional study on the COVID-19 vaccines' side effects.
The authors respond to the comments and suggested changes were done in the revised text.
The authors' research work fits in the journal's scope and currently, such research studies are important.
The authors address the comments well. New references were cited.
This paper shows interesting results, qualifies for publication in the Peer J, and can be accepted for publication.

Experimental design

no comment

Validity of the findings

no comment

Additional comments

The article is a significant study. The methodology is very well described; the authors discuss the data obtained through methods in the discussion part.
Add the new references that were cited to the reference list.
You did a good discussion of the literature in light of the existing body of knowledge. Good luck